# Experimental Measurements and Thermodynamic Optimization of the NaCl+RbCl Phase Diagram

**DOI:** 10.3390/ma15186411

**Published:** 2022-09-15

**Authors:** Zhangyang Kang, Maogang He, Guangxuan Lu

**Affiliations:** 1School of Environmental and Municipal Engineering, North China University of Water Resources and Electric Power, Zhengzhou 450046, China; 2Key Laboratory of Thermo-Fluid Science and Engineering of Ministry of Education, School of Energy and Power Engineering, Xi’an Jiaotong University, Xi’an 710049, China; 3Western Metal Materials Co., Ltd., Xi’an 710201, China

**Keywords:** differential scanning calorimetry, modified quasichemical model, phase diagram, thermodynamics, phase equilibria, activity coefficient

## Abstract

In this study, the phase diagram of the NaCl+RbCl binary system was measured using differential scanning calorimetry, and the measured molar percentages of the binary mixture RbCl ranged from 12 to 97 mol%, updating and extending the experimental data for this binary system. The liquid phase of this system was thermodynamically modeled using a modified quasichemical model, and a computer optimization program for evaluation of the experimental data on phase equilibria and other thermodynamic data was developed based on this model. All thermodynamic models of the NaCl+RbCl binary system were constructed; calculations of the phase diagram, enthalpy of mixing, and activity coefficient of NaCl of the NaCl+RbCl binary system were completed using the obtained model parameters; and the calculated eutectic point positions were *x*_RbCl_ = 0.567 and 550.2 °C. The calculated results were able to reproduce the various types of experimental data well, and the thermodynamic self-consistency between different types of experimental data was demonstrated.

## 1. Introduction

In recent years, four high-temperature molten salt materials, represented by nitrate, carbonate, chloride, and fluoride salts, have gained increased value in industrial applications with their excellent physicochemical properties. Compared with traditional heat transfer media (water and water vapor), molten salt materials have better characteristics, such as low vapor pressure, high boiling point, good thermal stability, relatively good thermal and electrical conductivities, higher specific heat, and high sensible and latent heat storage capacity. These four molten salts have not only led to the development of new technologies in energy fields such as nuclear power and solar energy but also play a key role in molten salt batteries, solid-state lasers, and material preparation.

The chemical stability of chloride salts is lower than that of fluoride salts but higher than that of nitrates and carbonates. Therefore, chlorinated salts are often used as heat transfer and heat storage media in advanced nuclear reactors, collector-type solar power, electrolytic production of rare earth metals, molten salt batteries, and other fields. Alkali metal chloride salts, such as LiCl+NaCl+MgCl_2_ and LiCl+KCl+MgCl_2_, can be used as primary coolants in molten salt fast reactor (MSFR) [1]. Chlorinated mixed salts (e.g., NaCl+KCl+ZnCl_2_) [2] are a high-temperature heat transfer fluid that can potentially be used in combination with advanced Brayton cycles for collector-type solar power systems. In the field of electrolytic reduction of rare earth metals, molten chloride LiCl+NaCl+KCl+YCl_3_ [3] is used as an electrolyte to prepare pure yttrium, and molten chloride LiCl+KCl+MgCl_2_+YCl_3_ is used as an electrolyte [4] to synthesize Mg+Li+Y alloys through electrolysis. The thermally activated cells currently being developed in the field of high-energy-density batteries are FeS_2_|KCl-LiCl|LiSi [5], CoS_2_|LiCl-LiBr-LiF|LiSi [6], Li|LiCl-KCl-RbCl-LiF|FeS_2_ [7], etc. Alkali metal halides containing the cation Li^+^ are suitable electrolyte materials because of their lower melting point (compared with alkali metal halides containing the cation Na^+^), lower vapor pressure, and high ionic conductivity.

Ternary chloride salts containing the binary system NaCl+RbCl and Li+ ions are a potential molten salt cell. The addition of rare earth metal elements to the binary system NaCl+RbCl will be used for the electrolytic reduction of rare earth metals. The experimental and theoretical study of the phase equilibrium properties of the NaCl+RbCl binary system described in this paper will help engineers to understand the thermodynamic properties of chlorides containing Na^+^ and Rb^+^ cations, and will help with the design of new molten salt materials in the future.

## 2. Experimental Methods

### 2.1. Differential Scanning Calorimetry

Differential scanning calorimetry (DSC), as a thermal analysis technique, achieves the equality of sample and reference temperatures at any moment and follows the procedure to raise and lower the temperature. Because the sample and reference require different heating amounts, there is a heating amount difference. After a DSC measurement, a DSC curve is plotted. It represents heat flow as a function of temperature or time. There are two types of DSC, the power compensated DSC and the heat flow DSC, and they use different measurement methods.

Power-compensated DSC adopts the “zero equilibrium” principle, which is obtained when the temperature difference Δ*T* between sample and reference is equal to 0. This principle should be permanently maintained during measurements. Power-compensated DSC consists of two separate heating control systems. Through precise control of the heating power, one system ensures that the reference and sample are heated or cooled at a uniform heating or cooling rate, and the other system ensures compliance with the zero equilibrium principle must, permanently providing Δ*T* = 0. The sample and reference furnaces are individually heated or cooled according to the same temperature–time program. During the cooling process, the compensation heating wire rapidly heats the reference, and during the heating process, the compensation heating wire starts to heat the specimen, always keeping the temperature difference Δ*T* equal to 0.During the experimental measurement, the difference between the power applied to the investigated sample and to the reference is measured as a function of temperature or time. The working principle of heat-flow DSC is similar to that of the power-compensated DSC. The difference between them is that the sample and reference are simultaneously heated on an electrically heating platform by a single heater. The heating platform is made of a material with good thermal conductivity, which allows for rapid heat transfer between the sample and reference. This high thermal conductivity material ensures that the temperature difference between the reference and sample is close to 0 when the thermal effect occurs.

Figure 1 shows a schematic diagram of the heat-flow-type and power-compensation-type DSC. Power-compensated DSC has separate heaters for the sample and reference. They are separately heated at the command of the power compensation system. In a heat-flow DSC, the sample and reference are in the same heating chamber and are placed on a heating platform with good thermal conductivity. The heater under the heating platform simultaneously heats the sample and the reference. Figure 2 depicts a schematic diagram of a typical heat-flow DSC. The sample and reference crucibles are placed on a copper conduction heating platform; two nickel–chromium alloy disks are placed underneath the platform, directly beneath the sample and reference. The thermocouple connection points for measuring temperature and temperature difference are connected under the nickel–chromium alloy disk.

### 2.2. DSC Curve Analysis

Currently, differential scanning calorimetry (DSC) is the most used method, and a large amount of experimental data on the phase diagrams of binary and multiple chloride systems have been accumulated through experiments. However, the main disagreement lies in how to analyze and determine the characteristic points in the phase diagram based on the obtained DSC curves. This has become an important part of experimental accuracy. The analysis of differential thermal analysis (DTA) curves of a large number of alloys by Boettinger et al. [8], which we also adopted in this study, is now commonly used internationally.

Essentially, power-compensated DSC is the real DSC. It can comply with the zero equilibrium principle and guarantee that the temperature difference between the reference and sample is equal to 0 at all times. The basic configurations of DTA and heat-flow-type DSC are the same. Their heating sources for controlling the temperatures of the reference and sample are different. The heating sources of DTA and heat-flow-type DSC are the furnace wall and the separate heater respectively. Both the DTA and the heat flow type DSC can quantitatively measure the heat of phase transition. Their differences mainly exist in sensitivity and accuracy. Therefore, the analytical methods of DTA and DSC curves are basically the same.

The ideal case of the DTA curve for the melting and solidification process of a pure substance is shown in Figure 3. The melting point of a pure substance can be determined by the extrapolation onset point of the heating peak and by the extrapolation termination point of the cooling peak. For binary molten salts or binary alloy materials, take the phase diagram of the A–B binary system in Figure 4 as an example. The phase diagram of the A–B binary system is a simple phase diagram with one eutectic point. The eutectic temperatures and liquidus temperatures can be determined by the onset point of the first endothermic peak and the peak point of the second endothermic peak in the endothermic process. They can also be determined using the termination points of the first and second exothermic peaks of the DTA curve during the exothermic process. This analytical method was proposed by Boettinger et al. [8] after analyzing the DTA curves of a large number of alloys. In this study, this method was used to analyze the measured DSC curves.

### 2.3. Experimental Instruments and Materials

The thermal analysis measurements completed in this study were performed using a synchronous thermal analyzer (TGA-DSC) (NETZSCH, instrument model: STA 449 F3 Jupiter, Selb, Germany). It allows the measurement of thermal effects at temperatures ranging from room temperature to 1450 °C. Pure NaCl and RbCl, with a purity of 99.99% and 99.95%, respectively, were purchased from Aladdin. The pure RbCl was slightly hygroscopic. In order to dry the pure RbCl, it was dried in a vacuum-drying oven at a temperature of 200 °C for 2 h. After completing drying, the pure RbCl was stored in a dry glove box. We prepared about 40 mg of the binary mixture and placed it in a platinum–rhodium crucible. Nitrogen was used as a protective gas, and the temperature rise and fall rate were set to 10 K·min^−1^. Two heating and cooling cycles were completed. The first cycle was performed to obtain a homogenized binary salt mixture. The DSC curves obtained from the first cycle were not used for analysis. The DSC curves obtained from the second heating and cooling cycle were mainly analyzed.

## 3. Experimental Results and Discussion

### 3.1. Experimental Results for Pure Substances

Figure 5 presents the DSC curves of pure NaCl and RbCl. As shown in Figure 5a, for pure NaCl, the melting temperatures obtained from the curve analysis are 803.4 °C (heating curve) and 800.1 °C (cooling curve). As shown in Figure 5b, for pure RbCl, the obtained melting temperatures are 721.7 °C (heating curve) and 719.2 °C (cooling curve). These values are listed in Table 1. It can be seen that the effect of supercooling on the melting point is negligible.

### 3.2. Experimental Results of Binary System

Figure 6 illustrates the experimentally obtained DSC curves for the NaCl+RbCl mixtures with different mole fractions *x*_RbCl_. Table 1 presents the analytically determined eutectic and liquidus temperatures. As shown in Figure 6, basically all of the cooling DSC curves contain two exothermic peaks, and some of the heating curves have only one heat endothermic peak. The eutectic temperature of the NaCl+RbCl binary system can be clearly reflected by the endothermic and exothermic peaks. The liquidus line can also be determined by the exothermic peaks. However, some endothermic peaks cannot reflect the liquidus temperature. The reason may be that the compositions of some NaCl+RbCl mixtures are very close to the eutectic composition, and the endothermic peaks of the eutectic reaction are too strong to cover up the endothermic peak for reflecting the liquidus lines in the heating process, such as the heating DSC curves with an *x*_RbCl_ equal to 52.5 and 57.0 mol%. It is also possible that anthropogenic or nonanthropogenic interference affected the formation of peaks, and a garbled signal appeared. The heating DSC curves for *x*_RbCl_ equal to 40.7 and 67.6 mol% reflect this situation.

As can be seen in Table 1, due to the effect of supercooling, in most cases, the phase transition temperature determined by the heating curve is higher than that determined by the cooling curve. Only when *x*_RbCl_= 30.6 mol%, its liquidus temperature determined by the heating curve is lower than that determined by the cooling curve. Therefore, for this mole fraction, only the liquidus temperature determined by the cooling curve was used in this study. For the two DSC curves that are very close to the pure RbCl, such as *x*_RbCl_= 96 and 97 mol%, the endothermic and exothermic peaks on the DSC curve reflecting the eutectic reaction become very small, especially on the cooling DSC curve, where the exothermic peak reflecting the eutectic reaction is almost invisible. This indicates that the eutectic reaction is no longer occurring here, so for the thermodynamic optimization in this study, we used only the liquidus temperatures.

### 3.3. Error Analysis of DSC Measurements

Errors in DSC measurement mainly come from the systematic error of DSC instruments and personal errors during the operating procedure (including sampling and data processing). The systematic error of DSC instruments includes thermal analysis accuracy and DSC curve analysis error. NETZSCH’s software can be used to mark the start point, peak, inflection point, and endpoint temperatures of peaks, and to perform automatic peak searches. The sources of personal errors mainly come from sample storage, dehumidification, weighing, and mixing.

## 4. Thermodynamic Optimization and Calphad Method

### 4.1. Thermodynamic Model

Table 2 presents all thermodynamic data ($\Delta H_{298.15\;K}^{0}$,$S_{298.15\;K}^{0}$, and *C_p_*(*T*)) for the condensed pure compounds of the NaCl+RbCl systems. The molar Gibbs free energy of a stoichiometric phase can be written as:(1)$$G(T)=\Delta H_{298.15\;K}^{0}-T\cdot S_{298.15\;K}^{0}+\int_{298.15\;K}^{T}C_{p}(T)dT-T\cdot \int_{298.15\;K}^{T}\left(\frac{C_{p}(T)}{T}\right)dT$$

The modified quasichemical model [10,11] was used to describe the liquid phase of the NaCl+RbCl system. The nonconfigurational Gibbs energy change Δ*g*_AB_ containing the parameters of this model is an important term in the expression for the Gibbs free energy of the solution. Δ*g*_AB_ can be expanded as a polynomial as:(2)$$\Delta g_{\mathrm{AB}}=\Delta g_{\mathrm{AB}}^{0}+\sum_{(i+j)\geq 1}g_{AB}^{ij}\chi_{\mathrm{AB}}^{i}\chi_{\mathrm{BA}}^{j}$$
where $\Delta g_{\mathrm{AB}}^{0}$ and $g_{AB}^{ij}$ are the parameters of the model that can be functions of temperature. These parameters can be determined by optimizing the phase equilibrium and thermodynamic data, which were measured by the experimental method.$\chi_{\mathrm{AB}}$ and $\chi_{\mathrm{BA}}$ are component variables, and they can be defined as:(3)$$\chi_{\mathrm{AB}}=\left(\frac{X_{\mathrm{AA}}}{X_{\mathrm{AA}}+X_{\mathrm{AB}}+X_{\mathrm{BB}}}\right)$$
(4)$$\chi_{\mathrm{BA}}=\left(\frac{X_{\mathrm{BB}}}{X_{\mathrm{AA}}+X_{\mathrm{AB}}+X_{\mathrm{BB}}}\right)$$
where *X*_AA_, *X*_AB_, and *X*_BB_ represent the pair fractions of cation–cation pairs.

There are two terminal solid solutions in the NaCl+RbCl system: the terminal solid solution *α* (near the pure NaCl) and the terminal solid solution *β* (near the pure RbCl). For the solid solution *α*, the molar fraction *x*_RbCl_ is very small, and its excess Gibbs energy expression can be written as:(5)$$\Delta_{xs}G^{\alpha }=x_{\mathrm{RbCl}}RT\ln\gamma_{\mathrm{RbCl}}≅x_{\mathrm{RbCl}}L^{\alpha }$$
where *L^α^* is a temperature-independent coefficient. Solid solution *β*, it can be considered that the Na+ cations enter into the interspaces of the RbCl crystal structure. Using the classical polynomial, its excess Gibbs energy expression can be written as:(6)$$\Delta_{xs}G^{\beta }=x_{\mathrm{NaCl}}x_{\mathrm{RbCl}}(a+bx_{\mathrm{NaCl}}+cx_{\mathrm{RbCl}})$$
where *a*, *b*, and *c* are temperature-independent coefficients. The values of *L^α^*, *a*, *b*, and *c* all need to be determined by thermodynamic optimization.

### 4.2. Calphad Method

The essence of the Calphad method is the computer coupling of phase diagrams and thermochemistry. A classic book describing the generation and application of the Calphad techniques was authored by Saunders and Miodownik [12]. The experience and skill for the thermodynamic optimization of phase diagrams were provided by Hari Kumar and Wollants [13]. After learning a great deal about the Calphad method from the above references, we developed a computer program to perform all the optimization and calculations in this work. The calculating flow of our computer program was based on the flowchart of the Calphad method in [13]. This computer program has been used to perform thermodynamic evaluation and optimization for some binary and ternary chloride molten salt systems [14,15,16]. In this program, the minimum Gibbs energy of the system was found through the algorithms of golden section search and parabolic interpolation, and the optimal model parameters were obtained by Nelder–Mead simplex algorithm [17].

### 4.3. Results and Discussion

In this study, all available phase equilibrium property data and thermodynamic data for the binary NaCl+RbCl system were carefully evaluated. The phase diagram data of the NaCl+RbCl system measured in this study were also simultaneously evaluated. These experimental data used in the thermodynamic optimization process were the activity of the NaCl of the NaCl+RbCl system [18], enthalpy of mixing [19], and phase diagram [20,21]. The optimization of the phase diagram and thermodynamic properties of the NaCl+RbCl system was performed using a computer program we developed. Firstly, the data of mixing enthalpy, activity, and eutectic point were used to determine the initial values of the model parameters. Secondly, all experimental data were used to optimize the model parameters.

Pelton and Flengas [20] measured the phase diagram of the NaCl+RbCl system using the step-cooling curve method in 1970. The molar percentages of RbCl of the compositions for binary mixtures they reported ranged from 10 to 90 mol%. The one reported eutectic point was *x*_RbCl_ = 0.56 at 550 °C. In this study, the phase diagram of the NaCl+RbCl system was also measured using the DSC method, and the molar percentages of RbCl ranged from 12 to 97 mol%. More data for the eutectic temperature are also given. Short and Roy [21] measured the solubility limits of the two terminal solid solutions in the NaCl+RbCl system by using the interdiffusion method. They found that, at 515 °C, the solubility limit of RbCl in solid solution α (where NaCl is the main body) is 0.8 mol%, and the solubility limit of NaCl in solid solution β (where RbCl is the main body) is 6 mol%. Figure 7 shows a comparison of the calculated phase diagram of the NaCl+RbCl system with the corresponding experimental data. The calculated values basically reproduced the experimental data. The calculated solubility lines for the two solid solutions below the eutectic temperature also reproduced the experimental data given by Short and Roy. The calculated eutectic points are *x*_RbCl_= 0.567 and 550.2 °C, which are also in better agreement with the results recommended by Pelton and Flengas.

Figure 8 and Figure 9 show the calculated enthalpy of mixing and activity of the liquid phase of the NaCl+RbCl system, respectively, in which we compared the corresponding experimental data. It can be seen that the calculated enthalpy of mixing of the liquid phase at 810 °C is in good agreement with the experimental data [19]. Only the calculated activity of NaCl somewhat deviates from the experimental data [18], and the average relative error between the calculated and experimental values is 2.86%.

The model parameters Δ*g*_AB_ of the modified quasichemical model in Equation (2) obtained by optimization are:(7)$$\begin{array}{l} \Delta g_{\mathrm{NaRb}/\mathrm{Cl}}=-1137.35+1.11T/K+(-34.22-0.75T/K)\chi_{\mathrm{NaRb}}\\ +(55.08+0.42T/K)\chi_{\mathrm{RbNa}}\;J/\mathrm{mol} \end{array}$$

For solid solution *α* (NaCl is the main body), the optimized excess Gibbs energy parameter $\Delta_{xs}G^{\alpha }$ in Equation (5) is obtained as:(8)$$\Delta_{xs}G_{\mathrm{NaRb}/\mathrm{Cl}}^{\alpha }=x_{\mathrm{RbCl}}RT\ln\gamma_{\mathrm{RbCl}}≅31078.51x_{\mathrm{RbCl}}\;J/\mathrm{mol}$$

For solid solution *β* (RbCl is the main body), the optimized excess Gibbs energy parameter $\Delta_{xs}G^{\beta }$ in Equation (6) is obtained as:(9)$$\Delta_{xs}G_{\mathrm{NaRb}/\mathrm{Cl}}^{\beta }=x_{\mathrm{NaCl}}x_{\mathrm{RbCl}}(9587.93+92749.24x_{\mathrm{NaCl}}+2014.85x_{\mathrm{RbCl}})\;J/\mathrm{mol}$$

## 5. Conclusions

In this study, differential scanning calorimetry analysis of the various compositions in the NaCl+RbCl system was performed. A thermodynamic data set for the NaCl+RbCl system was generated to reproduce all available experimental data. The following conclusions were obtained:

(1) The phase diagram of the NaCl+RbCl system was measured using DSC, and the measured molar percentages of RbCl of the binary mixture ranged from 12 to 97 mol%, which enriches the experimental data of the phase diagram of the NaCl+RbCl system.

(2) With the help of a computational program that we developed, all available experimental data on phase equilibria and other thermodynamic data, also including the measurements in this work, were evaluated. The obtained model parameters were used to complete the calculations of the phase diagram, enthalpy of mixing, and activity for the NaCl+RbCl system.

(3) The phase diagram measurements and thermodynamic optimization obtained in this study provide a foundation for the subsequent completion of the phase diagram measurements and thermodynamic optimization of the multicomponent system containing NaCl+RbCl.

## Figures and Tables

**Figure 1 materials-15-06411-f001:**
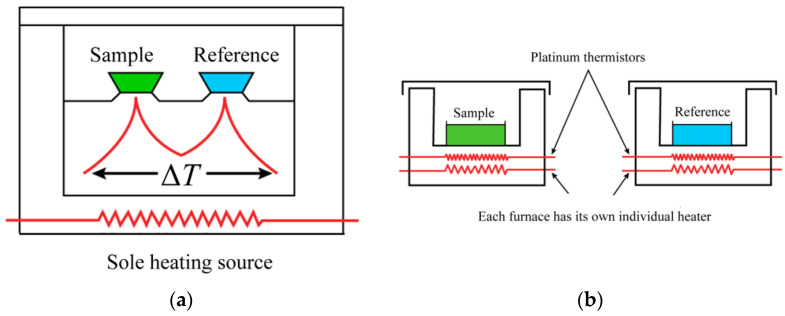
Schematic diagram of (**a**) heat-flow-type DSC; (**b**) power-compensation-type DSC.

**Figure 2 materials-15-06411-f002:**
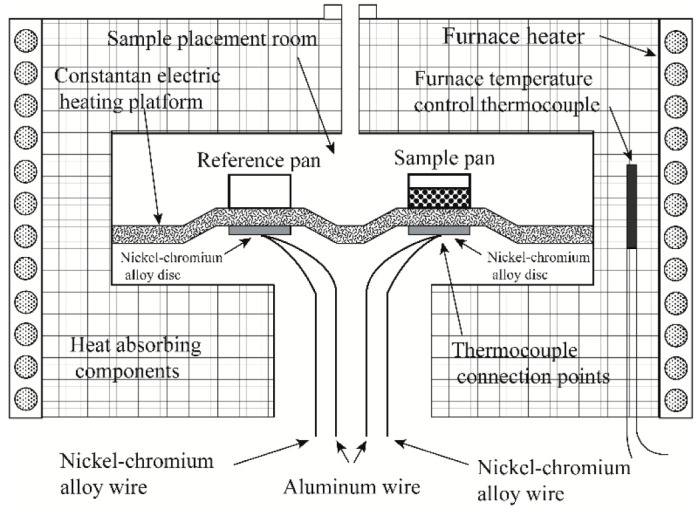
Schematic diagram of a typical heat-flow DSC.

**Figure 3 materials-15-06411-f003:**
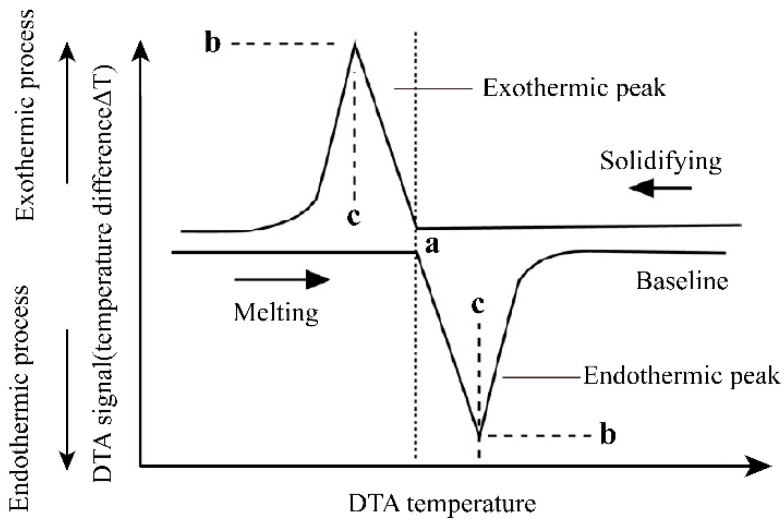
The ideal case of the DTA curve for melting and solidification process of pure substance. a—Peak onset temperature (pure substance melting point); b—peak point DTA signal; c—peak temperature.

**Figure 4 materials-15-06411-f004:**
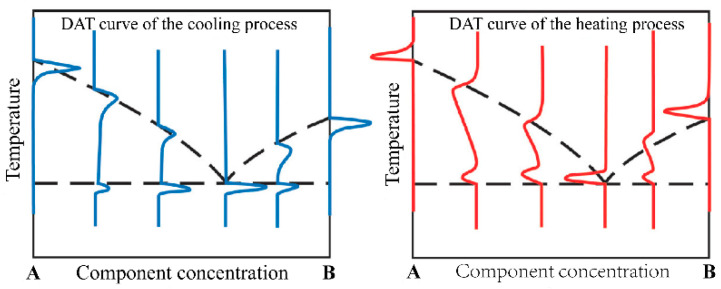
Eutectic points and liquid phase lines indicated by heating and cooling DTA curves in a simple eutectic phase diagram.

**Figure 5 materials-15-06411-f005:**
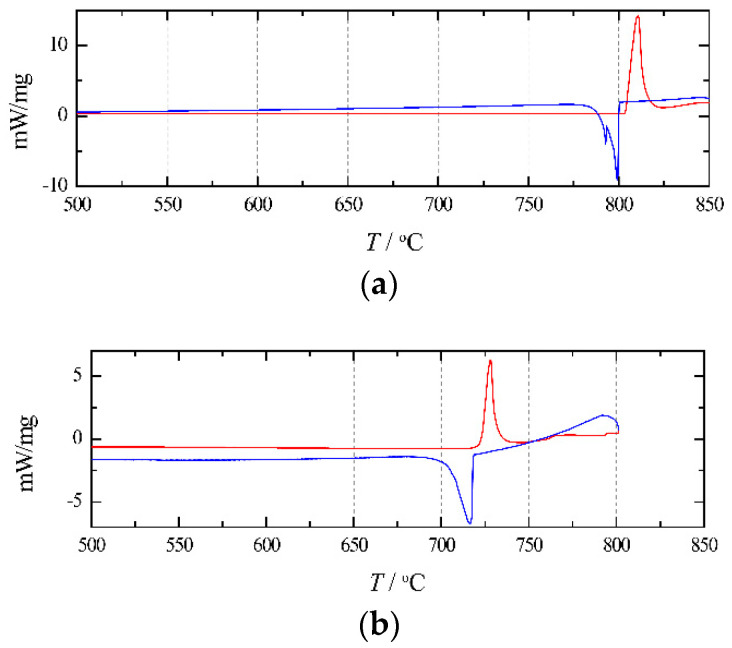
Heating (red/upper) and cooling (blue/lower) DSC-curves for (**a**) pure NaCl; (**b**) pure RbCl.

**Figure 6 materials-15-06411-f006:**
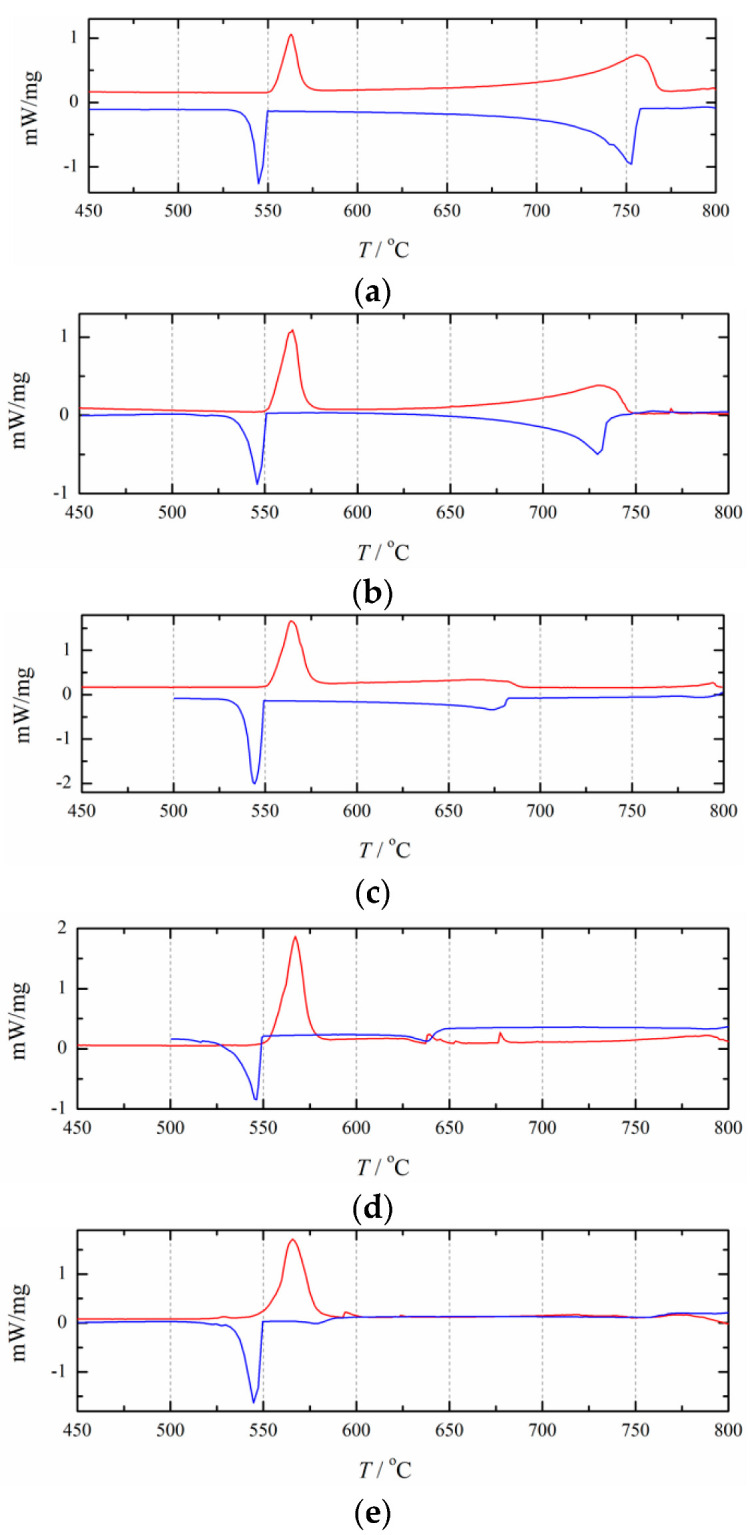
Heating (red/upper) and cooling (blue/lower) DSC-curves for 10 compositions in the (NaCl+RbCl) system:*x*_RbCl_ = (**a**) 12.1 mol%; (**b**) 18.2 mol%; (**c**) 30.6 mol%; (**d**) 40.7 mol%; (**e**) 52.5 mol%; (**f**) 57.0 mol%; (**g**) 67.7 mol%; (**h**) 78.6 mol%; (**i**) 96 mol%; (**j**) 97 mol%.

**Figure 7 materials-15-06411-f007:**
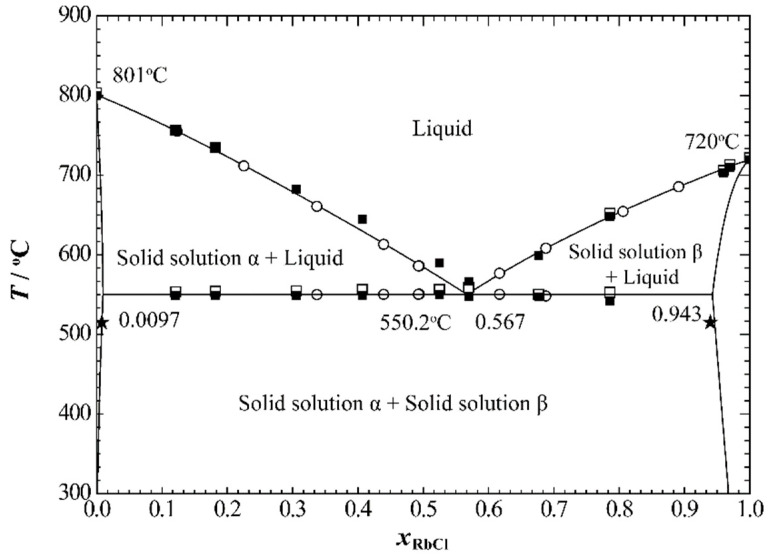
The calculated phase diagram of the NaCl+RbCl system at atmospheric pressure. □, Heating process; ■ cooling process phase diagram points measured in this study; (◦) Pelton and Flengas [20]; (★) Short and Roy [21].

**Figure 8 materials-15-06411-f008:**
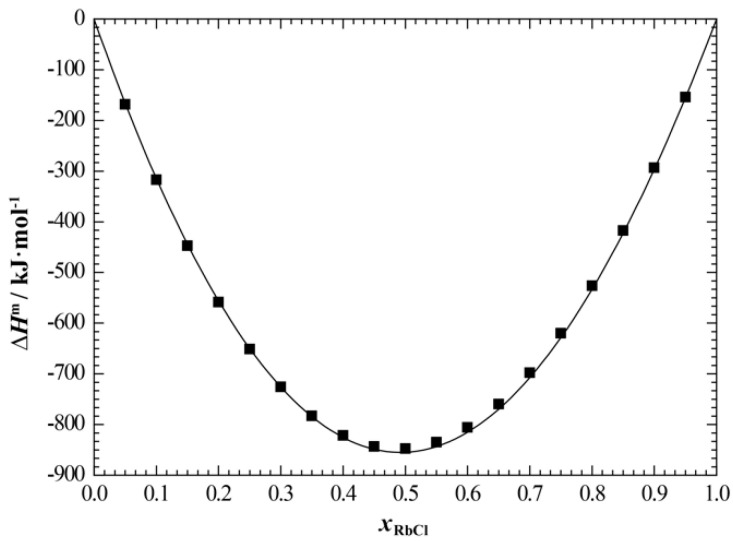
Calculated enthalpy of mixing of the liquid phase of NaCl+RbCl system (810 °C) at atmospheric pressure. ■, Hersh and Kleppa [19].

**Figure 9 materials-15-06411-f009:**
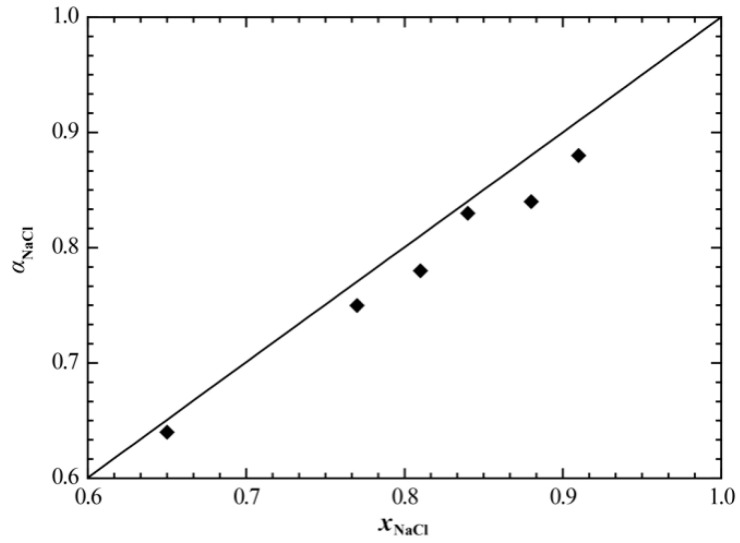
Calculated activity of NaCl in the liquid phase of the NaCl+RbCl system (865 °C). ♦, Topor et al. [18].

**Table 1 materials-15-06411-t001:** Eutectic temperature and liquidus temperatureof the NaCl+RbCl system determined from the DSC curve.

*x*_RbCl_ (mol%)	Liquidus Temperature (°C)		Solidus Temperature (°C)	
	Heating Curve	Cooling Curve	Heating Curve	Cooling Curve
0	803.4	800.1	-	-
12.1	756.3	756.0	553.4	549.0
18.2	734.4	734.1	554.2	549.3
30.6	665.6 *	682.0	554.4	549.0
40.7	-	644.3	556.1	549.3
52.5	-	589.4	556.3	549.4
57.0	-	566.3	558.8	547.5
67.7	-	598.8	550.2	546.9
78.6	652.0	647.8	552.7	541.8
96.0	706.0	702.5	550.6*	516.5 *
97.0	713.1	709.3	550.9*	565.7 *
100	721.7	719.2	-	-

Note: Experimental pressure was 96,994 Pa. * the phase inversion temperatures that were not used in this study.

**Table 2 materials-15-06411-t002:** The standard enthalpy ($\Delta H_{298.15\;K}^{0}$), absolute entropy ($S_{298.15\;K}^{0}$), and *C_p_*(*T*) at 298.15 K for pure NaCl and RbCl [9].

**Compounds**	Temperature Range (K)	$\Delta H_{298.15\;K}^{0}\;(J\cdot {\mathrm{mol}}^{-1})$	$S_{298.15\;K}^{0}\;(J\cdot K^{-1}\cdot {\mathrm{mol}}^{-1})$	C*_p_* (J·K^−1^·mol^−1^)
NaCl(s)	298.15–2000	−411,119.8	72.1322	45.9403 + 0.0163176(*T*/K)
NaCl(l)	298.15–1500	−394,956.0	76.0761	77.7638 + 0.0075312(*T*/K)
NaCl(l)	1500–2000	−390,090.1	84.5055	66.9440
RbCl(s)	298.15–2000	−430,533.6	91.6296	48.1160 + 0.0104182(*T*/K)
RbCl(l)	298.15–2000	−418,498.2	98.2792	64.0152

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
