# Peer review of "Experimental Measurements and Thermodynamic Optimization of the NaCl+RbCl Phase Diagram"

_materials, 2022, doi:10.3390/ma15186411_

Round 1
Reviewer 1 Report
Paper by Kang Zhangyang et al. described DSC measurements on NaCl-RbCl and thermodynamic assessment of data acquired by authors and known in literature. I did not find anything particularly wrong in the manuscript, and think it may be published. However, there are several points that make me very unhappy about the paper:
(a) I can not really understand how new (by authors) and numerous previous data on the NaCl-RbCl system are different. In other words, what is real scientific value and significance of the work?
(b) DSC method is known for many decades, it description given in text-book, and “Wikipedia-style” introduction in the method in the scientific paper is obviously not necessary. I am sorry if I missed something, but I got impression that 1/3 of the paper describe DSC as new method with diagrams and illustrations I learn as student 40 years ago.
(c) Authors describe results of physical measurements. It means that detail analysis of experimental and systematic uncertainties have to be presented, an all error-bars given in tables and figures.
Author Response
The comments from Reviewer 1
#1: I can not really understand how new (by authors) and numerous previous data on the NaCl-RbCl system are different. In other words, what is real scientific value and significance of the work?
Response: Thank you for your advice. What you say is not entirely true. Compared to previous studies, we are research work is not groundbreaking. But our work is also meaningful. It is mainly manifested in three aspects. First, the correctness of the previous experiments was verified by a different method than the previous ones. Second, we have a broader range of experiments compared to the previous experiments. Some phase diagram data close to the pure mass part were added. Third, all available thermodynamic data were evaluated with a Modified Quasichemical Model and a database for this binary system was constructed. This work provides a solid foundation for subsequent thermochemical calculations of multivariate systems containing the NaCl-RbCl system.
#2: DSC method is known for many decades, it description given in text-book, and “Wikipedia-style” introduction in the method in the scientific paper is obviously not necessary. I am sorry if I missed something, but I got impression that 1/3 of the paper describe DSC as new method with diagrams and illustrations I learn as student 40 years ago.
Response: Many thanks for this good suggestion. The description of DSC is indeed ambiguous, leading the reader to believe that this is a new technology. The DSC method is not a new method for measuring phase diagrams. 60 years ago Differential scanning calorimetry (DSC) was developed in 1962 by Perkin-Elmer employees Emmett Watson and Michael O’Neill. Moreover, this paper is too detailed in the introduction of differential scanning calorimetry technique. A detailed comparison of the differences between the heat flow type and the power compensated DSC is given.
Therefore, we have cut down some content and added some history of DSC. The specific description is like this: Differential scanning calorimetry (DSC) developed 60 years ago is a technique used to measure the difference in the heat flow rate of a sample and a reference over a controlled temperature range. These measurements are used to create phase diagrams and gather thermoanalytical information such as transition temperatures and enthalpies.
#3: Authors describe results of physical measurements. It means that detail analysis of experimental and systematic uncertainties have to be presented, an all error-bars given in tables and figures.
Response:
Thank you for your advice. The detail analysis of experimental and systematic uncertainties have been given in our work. But the error-bars are hard for me to give in tables and figures. The main reason is that the amount of our experimental mixture is very small, only 20-40 mg. They can't be physically mixed after they're weighed, they can only be mixed in heat. A mixed sample can be prepared only once.

Reviewer 2 Report
See attachment

Author Response
The comments from Reviewer 2
#1: The accuracy of all measured values should be mandatory presented in text.
Response: Thank you for your advice. The detail analysis of experimental and systematic uncertainties have been given in our work.
#2: The pressure, for which the data of Table 1 is obtained, should be also pointed out. The same concerned Figs. 7 and 8.
Response: Thank you for your advice. The experiments are conducted at local atmospheric pressure. The average atmospheric pressure is 96980 Pa measured by atmospheric manometer (type: testo 511). Therefore, in Table 1 the pressure is 96980 Pa.
All available thermodynamic data were evaluated with a Modified Quasichemical Model and a database for this binary system was constructed. These experimental data were basically measured at a standard atmospheric pressure. So Figure 7 and Figure 8 are expressed as “The calculated phase diagram of the NaCl-RbCl system at atmospheric pressure” and “Calculated enthalpy of mixing of the liquid phase of NaCl-RbCl system (810 oC) at atmospheric pressure”
#3: Present the decoding of DTA when this abbreviation is met for the first time (i. e. P. 3, line 108).
Response: Thank you for pointing out the mistake. The sentence has been modified as follows. “The analysis of differential thermal analysis (DTA) curves of a large number of alloys by Boettinger et al.”
#4: Besides, correct the following text (P. 1 line 30): Change "possess good heat conduction and electrical conductivity" by "possess relatively good thermal and electrical conductivities"
Response: Thank you for pointing out the mistake. The sentence has been modified as follows. “Compared with traditional heat transfer media (water and water vapor), molten salt materials have more excellent characteristics, such as: low vapor pressure, high boiling point, good thermal stability, possess relatively good thermal and electrical conductivities, higher specific heat, and high sensible and latent heat storage capacity.”

Round 2
Reviewer 1 Report
Authors' response is adequate